# Relationship between parents' mental disorders and socioeconomic status and offspring's psychopathology: A cross-sectional study

Alba Oliver-Parra[1,2], Albert Dalmau-Bueno[1], Dolores Ruiz-Muñoz[1], Anna García-Altés[1]*

**1** Agència de Qualitat i Avaluació Sanitàries de Catalunya (AQuAS), Barcelona, Spain, **2** Universitat Pompeu Fabra, Barcelona, Spain

* agarciaaltes@gencat.cat

**Data Availability Statement:** All relevant data are within the paper and its Supporting Information files. Individual level data cannot be shared publicly

## Abstract

Mental disorders (MD) are one of the main causes of the disease burden worldwide. Associations between socioeconomic status (SES) and presence of MD in parents have been related with increased odds of MD in offspring. However, there is a lack of population-based research in this field. The aim of the present study was to examine together the relationship between the presence of MD in children, and the SES and presence of MD in their parents, in a whole of population data. A gender approach was undertaken aiming to discern how these variables influence children's mental health when related with the father and the mother. Using administrative individual data from the National Health System, a retrospective cross-sectional study was conducted. The entire children population aged 6 to 15 resident in Catalonia in 2017 was examined. A logistic regression model was performed. Low SES was associated with increased odds of children's MD. Offspring of a parent with MD were at more risk of presenting MD than offspring of parents without these problems. Although these associations were consistent for both boys and girls when looking at the father's or mother's SES and MDs, the mother's SES and MDs showed a higher association with the offspring's MDs than the father's. Lowest associations, found for boys when looking at the father's SES and MDs, were: OR of 1.21, 95%CI 1.16 to 1.27 for lowest SES, and OR of 1.66, 95%CI 1.61 to 1.70 for parental MDs. Children's familiar environment, which includes SES and mental health of parents, plays an important role in their mental health. Socially constructed gender roles interfere with SES and parent's MD. These findings support the relevance of examining MD and its risk factors within a gender approach.

## Introduction

Mental health is an essential component of health [1], and mental disorders (MD) are one of the main causes of the disease burden worldwide [2]. In the case of children, poor mental health also impacts negatively the capacities and opportunities they will have in their adult life [3].

because of confidentiality. You can check up-to-date regulations at http://aquas.gencat.cat/ca/ambits/analitica-dades/padris/ and address any inquiries to AQuAS director, Dr. Cesar Velasco (cesarvelasco@gencat.cat).

**Funding:** The author(s) received no specific funding for this work.

**Competing interests:** The authors have declared that no competing interests exist.

**Abbreviations:** MD, Mental disorders; SES, Socioeconomic status; HCH, Health card holder.

Worldwide, it is estimated that approximately 10–20% of children and adolescents have mental health problems [4]. The disorders they suffer the most are those of an emotional and mood type and those of behaviour or hyperactivity [5]. These problems in children and adolescents are associated with difficulties in emotional and intellectual development [6] and thus, becoming problematic in adulthood. First onset of MD usually occurs in childhood or adolescence [7] and it is known that half of all MD in adulthood have already started by the age of 14 [8].

Developmental psychopathology includes consideration of sex differences and, especially, differences in the socialization of males and females. Both symptomatology and behaviour patterns predictive of later MD differ for males and females [9], placing gender as a critical determinant of mental health. Although there do not appear to be gender differences in the overall prevalence of MD [10], significant differences in the patterns and symptoms of the MD have been reported: women and girls show higher rates and lifetime risk of mood or internalizing disorders, while men and boys have a higher prevalence of behavioural or externalizing disorders [5, 11–13].

This general pattern can be seen throughout all life stages, but some differences across age groups have also been reported. Studies find that in childhood, there is a higher prevalence of conduct disorders among boys than among girls. During adolescence, girls have a much higher prevalence of depression and eating disorders than boys, while boys present more anger problems and high-risk behaviours. The median age of onset has been found much earlier for anxiety disorders and impulse-control disorders than for substance use disorders and mood disorders [7, 14]. There have also been found gender differences in the age of onset of severe MD such as schizophrenia and bipolar depression, with men typically having an earlier onset of schizophrenia, while women being more likely to exhibit serious forms of bipolar depression [10].

There is evidence for genetic susceptibility underlying the development of MD, but it has also been found that adverse environmental exposures also play an important role [15–17]. Epigenetic mechanisms have been suggested by recent research as possible pathways through which the environment interacts with genes and produces biological responses that seem to have a role in the onset and maintaining of psychopathology [18–24]. These gene-environment interactions indicate that genetic influences on the risk of children's psychopathology are moderated by environmental factors. Although the environment affects us throughout life, its effects are especially important during the sensitive periods of biological and brain development, which begin in the prenatal period and continue through childhood and adolescence [25].

During childhood and until the conclusion of adolescence, important biological, psychological and emotional changes occur in human beings. During this stage, the key dimensions of health are developed: the physical, cognitive and psychosocial dimensions [26]. While there is no uniformity in the terminology used to designate the stages of childhood nor the age ranges [27–29], it is well established that development throughout the different periods of childhood influence mental health outcomes across the lifespan: the prenatal period and early childhood -approximate age range: until 5 years of age- [30], middle childhood -approximate age range: 5 to 12 or so years of age- [31, 32] and also adolescence—approximate age range: 12 or so to 20 years of age, divided in early (11–14), middle (14–17) and late adolescence (17 and up)- [33–36]. During these life stages there is a greater vulnerability to the characteristics of the environment [30, 37–39] due to the high malleability of biological systems [40].

Children are strongly influenced by their environments, especially by their family unit. At this stage in life, the family is the main influence on the child's development, and the most influential characteristics of the family environment are its economic and social resources, including family members' health, especially early in this period [30] but also in late childhood and adolescence [41, 42]. Different theories in developmental psychology conceive the

interaction between the caregiver and the child as crucial to psychological outcomes. Following such theories, the risk of MD in children can be seen as a result of complex interactions across multiple levels of analysis from a child's genetic predispositions to environmental exposures such as their family conditions [41, 43].

There has been consistent evidence to demonstrate that parental MD affects almost every aspect of child development [44] and negatively affects the mental and emotional health of children [6]. Children of parents with MD present a greater risk of psychological problems [45–48]. Previous research indicates that for mothers, the association between parental and child psychopathology is specific, whereas for fathers it is non-specific: mothers' internalizing problems have been associated to child internalizing problems and the same applied for externalizing problems [49]. Furthermore, existing research, albeit limited, suggests that maternal intergenerational transmission of MD is particularly strong [49–54].

In addition, in situations of economic difficulty, there is a higher prevalence of MD in both children and adults [55, 56] and parents with mental health problems are at greater risk of socioeconomic disadvantage [44]. Adversity during childhood, including the presence of parental MD and economic difficulties, is associated with greater MD both in childhood and adulthood [57–60].

Since having a low socioeconomic status (SES) is a risk factor for suffering MD, children who grow up in deprived family environments are affected by both, the low SES and the greater likelihood that their parents will be suffering from MD. This places children on a disadvantaged situation, which will affect their present and future life trajectories, contributing to perpetuate inequalities.

Most of the previous studies that have examined the relationship between parental SES and mental disorders and children's mental disorders, have used representative samples rather than the whole of population data. The aim of this study is to analyse, within a gender approach, the association between parent's MD and SES and the prevalence of MD in their offspring using data from population registers of Catalonia. The specific objectives include the study of: 1) the association between parental MD and offspring's MD; 2) the association between parental SES and offspring's MD; and 3) the role played by the gender of both the parent and their offspring in these associations. We hypothesized that MD would be more prevalent in children with a parent with MD and with low SES. We also hypothesized that these associations would be higher in the case of maternal MD and low SES because socially constructed gender roles often place the woman as the principal caregiver. Finally, we hypothesized that there could be differences in the associations between parental characteristics and children's psychological outcome regarding the gender of the child.

## Materials and methods

### Ethics approval

This study adheres to the PADRIS Program. This Program is a data analytics program for health research and innovation of the Catalan healthcare administration. It has the mission of making available to the scientific community the related health data to promote research, innovation and evaluation in health through access to the reuse and crossing of health data generated by the comprehensive National Health System of Catalonia. It ensures the use of the data goes in accordance with the legal and regulatory framework, the ethical principles and transparency towards the citizens of the program. We have had access to the data as workers within the Catalan healthcare administration under the fulfilment of the established criteria to guarantee the respect to the ethical, security and risk analysis principles of the PADRIS Program, including the anonymization of all data made available to researchers. Thus, the study

did not involve any data collection, requiring neither human participants nor patient consent. For that reason, and due to the use of existing anonymised data for research, the study was exempt from institutional review committee approval. It is the standard way of proceeding in the healthcare administration to reuse the data provided by the healthcare registers for scientific purposes and to systematically check health outcomes of our population in our context.

## Study design and setting

A retrospective cross-sectional study with data from 2017 was conducted to examine the relationship between the presence of MD in children aged 6 to 15 resident in Catalonia, the presence of MD in one of their parents, and their SES.

Data was collected from three databases of the Catalan National Health System's administrative registries.

The Central Registry of Insured Persons (RCA) provided the identification of CatSalut insurers, their sociodemographic characteristics, the linkage between children and their health cardholder (HCH) and an individual variable of SES, which corresponds to the health cardholder's. Only a legal guardian of the children can legally be their HCH [61] which, in the vast majority of cases, will be a parent. Thus, from now on, the study will be referring to them as parents.

The Minimum Basic Data Set provided the records of all the contacts made by the target population with the public health care system for Primary Care, Hospitalization, Emergency and Mental Health Centers. Finally, the Pharmacy Billing Database provided information on the pharmaceutical dispensation of mental health drugs.

The RCA data was linked with the different registers of the Minimum Basic Data Set and the Pharmacy Billing Database by means of an anonymized individual code, obtained from the personal identification code.

## Study population

According to the official population cut of the RCA from January 1, 2017, the insured population of Catalonia, which during that year was between 6 and 15 years old, comprised 1,048,576 children.

There are several reasons for not including children less than six. First, there has been controversy about the validity of diagnosis of MD in very young children aged 2 to 5 years [5]. Second, children until six are visited for MD in centers outside the Catalan NHS (CDIAP), managed by the social services.

Some exclusion criteria were applied to recruit the target population. Individuals who could not be assigned a reliable SES were excluded (less than 0,01%), containing individuals affected by the toxic syndrome and disabled LISMI. Individuals who could not be linked to a HCH were also excluded (10,8%). This exclusion criterion comprised minors not classified as beneficiaries of any HCH, minors classified as beneficiaries but who did not provide the HCH's identifier, and minors guarded by the administration.

After the exclusion criteria, the final population included 934,948 children. However, missing data represented 1.03% of the target population (N = 9.669), due to the absence of information regarding nationality of the parent. These children were removed from the final sample, which included 925,279 children (Fig 1).

## Measures

The primary outcome was the presence of MD in children, which was constructed by crossing the different records of (1) use of health care services and (2) consumption of psychotropic

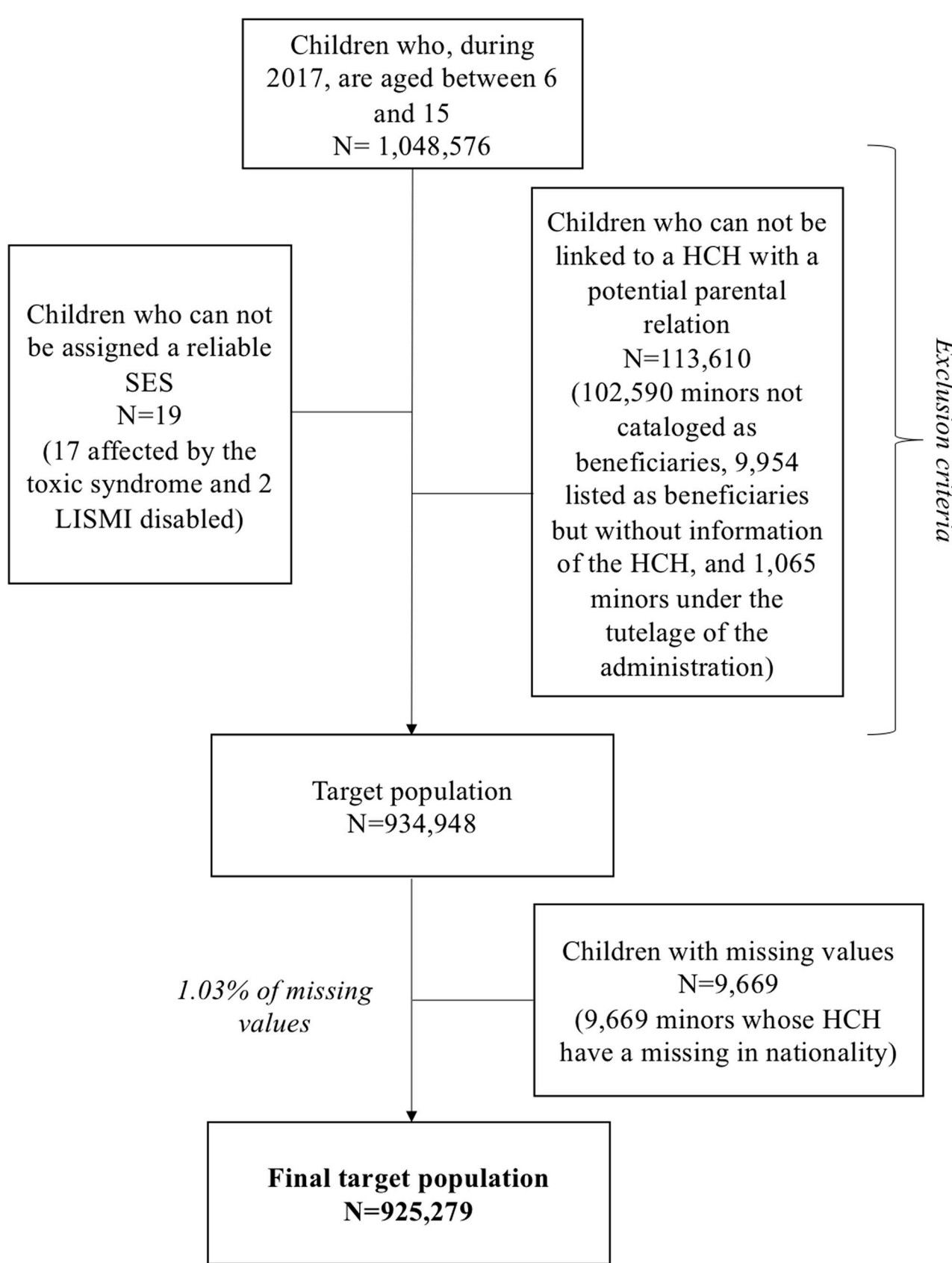

**Fig 1. Flow chart of the study population.**

drugs. The original variables were all dichotomous and indicated usage (yes or no). The use of the health care services was defined as follows: all the codes from the Clinical Classifications Software (CCS) considered within the large classification group number 5 of Mental Health of the ICD-9-CM (International Classification of Diseases- Clinical Modification) were taken into account. This group includes the following subgroups, with their respective codes: Psychosis (290–299); Neurotic disorders, personality disorders and other non-psychotic mental disorders (300–316); and Intellectual disabilities (317–319). All these variables regarding the use of public health care services and drug prescription, were summarized in a single dichotomous variable of presence or non-presence of MD, being the presence a usage in any of the original variables.

To address the role of gender, sex of both the children and their parents were treated as stratification variables. Age of the children was divided into two age groups with a 5-year interval in order to adjust for the different developmental stages: 6 to 10 and 11 to 15. Nationality of the parent was dichotomized into locals or foreigners. Age and nationality were treated as potential confounders.

The exposure variable of parent's SES was extracted from the sections of income and employment and social situation displayed by the pharmaceutical co-payment [62]. This variable is grouped into four categories: disadvantaged population (grouping all people who received an economical subsidy and those exempt from payment); income of less than 18,000 € per year; between 18,000 and 100,000 € per year; and more than 100,000 € per year.

The exposure to MD in parents was constructed following the same principles as in children.

## Statistical analysis

A descriptive analysis of the child population for all the variables of interest was carried out. The distribution of presenting MD in boys and girls according to age, parent's nationality, parent's SES, and whether or not their parents had MD, was examined.

The bivariate analysis explored the relationship between the independent variables and the outcome by means of the Chi-square test.

Finally, logistic regression models were constructed to assess the association between the different independent variables with the presence of MD in boys and girls. All the analysis was stratified by the sex of both, the descendant and the parent. The adjustment variables were age, nationality of the parent, SES of the parent and the presence of MD in the parent. Three sets of logistic regression models were fit to test the hypotheses. Model 1 assessed whether a lower SES increased the risk of MD in children. Model 2 assessed whether the risk for MD was greater among children whose parent had MD. In Model 3, both variables were included. All models were adjusted for potential confounders, including age and parent's nationality.

Statistical software Stata 14.2 was used for all analysis.

## Results

The study included 925,279 children who were between 6 and 15 during 2017, 476,304 boys (51.48%) and 448,975 girls (48.52%). The mean age of this population was 10.45 years old (SD = 3.14). Main characteristics of this population for the interest variables are described in Table 1.

MD were more common among boys (12,74% and 13,60% for those with the father as HCH and those with the mother as HCH, respectively) than girls (8,95% and 9,41% for those with the father as HCH and those with the mother as HCH, respectively). In contrast, there were more MD among mothers (27,38%) than fathers (20,75%).

**Table 1. Characteristics of children by sex.**

| | Boys (n(%)) | | Girls (n(%)) | |
|---|---|---|---|---|
| | **n** | **%** | **n** | **%** |
| **Father as health card holder** | 240,261 (51.61) | | 225,280 (48.39) | |
| **Age** | | | | |
| 6–10 | 121,994 | 50.78 | 114,234 | 50.71 |
| 11–15 | 118,267 | 49.22 | 111,046 | 49.29 |
| **Father's nationality** | | | | |
| Locals | 181,006 | 75.34 | 170,787 | 75.81 |
| Foreigners | 59,255 | 24.66 | 54,493 | 24.19 |
| **Father's socioeconomic status** | | | | |
| Disadvantaged population | 15,485 | 6.45 | 14,615 | 6.49 |
| Annual income <18.000 | 131,281 | 54.64 | 121,407 | 53.89 |
| Annual income 18.000–100.000 | 89,517 | 37.26 | 85,406 | 37.91 |
| Annual income >100.000 | 3,978 | 1.66 | 3,852 | 1.71 |
| **Mental disorders in the father** | | | | |
| No | 190,314 | 79.21 | 178,606 | 79.28 |
| Yes | 49,947 | 20.79 | 46,674 | 20.72 |
| **Mental disorders** | | | | |
| No | 209,663 | 87.26 | 205,114 | 91.05 |
| Yes | 30,598 | 12.74 | 20,166 | 8.95 |
| **Mother as health card holder** | 236,043 (51.34) | | 223,695 (48.66) | |
| **Age** | | | | |
| 6–10 | 120,362 | 50.99 | 112,961 | 50.50 |
| 11–15 | 115,681 | 49.01 | 110,734 | 49.50 |
| **Mother's nationality** | | | | |
| Locals | 209,023 | 88.55 | 198,324 | 88.66 |
| Foreigners | 27,020 | 11.45 | 25,371 | 11.34 |
| **Mother's socioeconomic status** | | | | |
| Disadvantaged population | 11,129 | 4.71 | 10,544 | 4.71 |
| Annual income <18.000 | 135,768 | 57.52 | 129,190 | 57.75 |
| Annual income 18.000–100.000 | 87,277 | 36.98 | 82,225 | 36.76 |
| Annual income >100.000 | 1,869 | 0.79 | 1,736 | 0.78 |
| **Mental disorders in the mother** | | | | |
| No | 171,298 | 72.57 | 162,586 | 72.68 |
| Yes | 64,745 | 27.43 | 61,109 | 27.32 |
| **Mental disorders** | | | | |
| No | 203,947 | 86.40 | 202,653 | 90.59 |
| Yes | 32,096 | 13.60 | 21,042 | 9.41 |

Disadvantaged population: People who received an economical subsidy and those exempt from pharmaceutical co-payment.

As seen in Table 2, the prevalence of MD was slightly higher among the oldest group of age for girls (i.e. 10.24%, 95%CI 10.06–10.42 for girls with the mother as HCH) and the youngest group of age for boys with the father as HCH (13.08%, 95%CI 12.89–13.27). For boys with a mother as a HCH there were no differences between ages. Locals presented higher rates of MD than foreigners did. These findings support including both variables as potential confounders.

SES showed the highest prevalence of MD in children from the lowest SES (up to 21.15%, 95%CI 20.40–21.92 for boys with the mother as HCH) and the lowest prevalence in the more

**Table 2. Prevalence of mental disorders for the different independent variables among children by sex.**

| | Boys (n(%)) | | | | Girls (n(%)) | | | |
|---|---|---|---|---|---|---|---|---|
| | **n** | **%** | **95%CI** | **p-value** | **n** | **%** | **95%CI** | **p-value** |
| **Father as health card holder** | 30,598 (12.74) | | | | 20,166 (8.95) | | | |
| **Age** | | | | | | | | |
| 6–10 | 15,961 | 13.08 | 12.89 to 13.27 | <0.001 | 9,462 | 8.28 | 8.12 to 8.44 | <0.001 |
| 11–15 | 14,637 | 12.38 | 12.19 to 12.57 | | 10,704 | 9.64 | 9.47 to 9.81 | |
| **Father's nationality** | | | | | | | | |
| Locals | 24,973 | 13.80 | 13.64 to 13.96 | <0.001 | 16,663 | 9.76 | 9.62 to 9.90 | <0.001 |
| Foreigners | 5,625 | 9.49 | 9.26 to 9.73 | | 3,503 | 6.43 | 6.22 to 6.64 | |
| **Father's socioeconomic status** | | | | | | | | |
| Disadvantaged population | 2,292 | 14.80 | 14.25 to 15.37 | <0.001 | 1,519 | 10.39 | 9.90 to 10.90 | <0.001 |
| Annual income <18.000 | 16,495 | 12.56 | 12.39 to 12.75 | | 10,945 | 9.02 | 8.85 to 9.18 | |
| Annual income 18.000–100.000 | 11,534 | 12.88 | 12.67 to 13.11 | | 7,524 | 8.81 | 8.62 to 9.00 | |
| Annual income >100.000 | 277 | 6.96 | 6.19 to 7.80 | | 178 | 4.62 | 3.98 to 5.33 | |
| **Mental disorders in the father** | | | | | | | | |
| No | 21,594 | 11.35 | 11.20 to 11.49 | <0.001 | 13,957 | 7.81 | 7.69 to 7.94 | <0.001 |
| Yes | 9,004 | 18.03 | 17.69 to 18.37 | | 6,209 | 13.30 | 13.00 to 13.61 | |
| **Mother as health card holder** | 32,096 (13.60) | | | | 21,042 (9.41) | | | |
| **Age** | | | | | | | | |
| 6–10 | 16,361 | 13.59 | 13.40 to 13.79 | 0.950 | 9,706 | 8.59 | 8.43 to 8.76 | <0.001 |
| 11–15 | 15,735 | 13.60 | 13.40 to 13.80 | | 11,336 | 10.24 | 10.06 to 10.42 | |
| **Mother's nationality** | | | | | | | | |
| Locals | 29,379 | 14.06 | 13.91 to 14.21 | <0.001 | 19,150 | 9.66 | 9.53 to 9.79 | <0.001 |
| Foreigners | 2,717 | 10.06 | 9.70 to 10.42 | | 1,892 | 7.46 | 7.14 to 7.79 | |
| **Mother's socioeconomic status** | | | | | | | | |
| Disadvantaged population | 2,354 | 21.15 | 20.40 to 21.92 | <0.001 | 1,670 | 15.84 | 15.15 to 16.55 | <0.001 |
| Annual income <18.000 | 19,971 | 14.71 | 14.52 to 14.90 | | 13,377 | 10.35 | 10.19 to 10.52 | |
| Annual income 18.000–100.000 | 9,654 | 11.06 | 10.85 to 11.27 | | 5,926 | 7.21 | 7.03 to 7.39 | |
| Annual income >100.000 | 117 | 6.26 | 5.20 to 7.46 | | 69 | 3.97 | 3.11 to 5.00 | |
| **Mental disorders in the mother** | | | | | | | | |
| No | 18,186 | 10.62 | 10.47 to 10.76 | <0.001 | 11,560 | 7.11 | 6.99 to 7.24 | <0.001 |
| Yes | 13,910 | 21.48 | 21.17 to 21.80 | | 9,482 | 15.52 | 15.23 to 15.81 | |

CI: Confidence Interval.

p-value from Chi-square test comparing the presence and absence of mental disorders in children.

Disadvantaged population: People who received an economical subsidy and those exempt from pharmaceutical co-payment.

privileged SES in all groups. As for the presence of MD in the parent, higher rates of MD among children were found when the parent had a MD himself (up to 21.48%, 95%CI 20.17–21.80 for boys with the mother as HCH).

Table 3 presents the results of crude and adjusted analyses stratified by sex of the children and of the parent. Boys aged between 6 and 10 were at more risk of presenting a MD than those of older ages. On the contrary, for girls, the highest risk was found in the oldest group (11 to 15). Having a foreigner parent was always a protective factor.

There was an association between the presence of MD in the parent and MD in the children. Also, between SES of the parent and MD in the children. These associations persisted after controlling for all variables. Both in boys and girls, the highest risks of MD were observed in those with a mother presenting MD (OR 2.17, 95%CI 2.12–2.23 and OR 2.23, 95%CI 2.17–

**Table 3. Risk of mental disorders among children associated with the parental socioeconomic status and presence of mental disorders.**

| | Boys (n(%)) | | | | | | | | Girls (n(%)) | | | | | | | |
|---|---|---|---|---|---|---|---|---|---|---|---|---|---|---|---|---|
| | Crude odds ratio | | Model 1 | | Model 2 | | Model 3 | | Crude odds ratio | | Model 1 | | Model 2 | | Model 3 | |
| | ORc | 95%CI | ORa | 95%CI | ORa | 95%CI | ORa | 95%CI | ORc | 95%CI | ORa | 95%CI | ORa | 95%CI | ORa | 95%CI |
| **Father as health card holder** | 240,261 (51.61) | | | | | | | | 225,280 (48.39) | | | | | | | |
| **Age** | | | | | | | | | | | | | | | | |
| 6–10 | 1 | | 1 | | 1 | | 1 | | 1 | | 1 | | 1 | | 1 | |
| 11–15 | 0.94 | 0.92 to 0.96** | 0.91 | 0.89 to 0.93** | 0.90 | 0.87 to 0.92** | 0.90 | 0.87 to 0.92** | 1.18 | 1.15 to 1.22** | 1.14 | 1.11 to 1.18** | 1.13 | 1.09 to 1.16** | 1.12 | 1.09 to 1.16** |
| **Father's nationality** | | | | | | | | | | | | | | | | |
| Locals | 1 | | 1 | | 1 | | 1 | | 1 | | 1 | | 1 | | 1 | |
| Foreigners | 0.66 | 0.64 to 0.68** | 0.61 | 0.59 to 0.63** | 0.67 | 0.65 to 0.69** | 0.64 | 0.62 to 0.66** | 0.64 | 0.61 to 0.66** | 0.59 | 0.57 to 0.62** | 0.67 | 0.64 to 0.70** | 0.63 | 0.60 to 0.65** |
| **Father's socioeconomic status** | | | | | | | | | | | | | | | | |
| Disadvantaged population | 1.21 | 1.15 to 1.27** | 1.27 | 1.21 to 1.33** | | | 1.21 | 1.16 to 1.27** | 1.17 | 1.11 to 1.24** | 1.24 | 1.17 to 1.31** | | | 1.17 | 1.11 to 1.24** |
| Annual income <18.000 | 1 | | 1 | | | | 1 | | 1 | | 1 | | | | 1 | |
| Annual income 18.000–100.000 | 1.03 | 1.00 to 1.06* | 0.91 | 0.88 to 0.93** | | | 0.93 | 0.90 to 0.95** | 0.98 | 0.95 to 1.01 | 0.86 | 0.83 to 0.88** | | | 0.88 | 0.85 to 0.91** |
| Annual income >100.000 | 0.52 | 0.46 to 0.59** | 0.47 | 0.42 to 0.54** | | | 0.52 | 0.46 to 0.59** | 0.49 | 0.42 to 0.57** | 0.44 | 0.37 to 0.51** | | | 0.48 | 0.41 to 0.55** |
| **Mental disorders in the father** | | | | | | | | | | | | | | | | |
| No | 1 | | | | 1 | | 1 | | 1 | | | | 1 | | 1 | |
| Yes | 1.72 | 1.67 to 1.77** | | | 1.69 | 1.64 to 1.73** | 1.66 | 1.61 to 1.70** | 1.81 | 1.75 to 1.87** | | | 1.76 | 1.70 to 1.81** | 1.72 | 1.66 to 1.77** |
| **Mother as health card holder** | 236,043 (51.34) | | | | | | | | 223,695 (48.66) | | | | | | | |
| **Age** | | | | | | | | | | | | | | | | |
| 6–10 | 1 | | 1 | | 1 | | 1 | | 1 | | 1 | | 1 | | 1 | |
| 11–15 | 1.00 | 0.98 to 1.02 | 1.00 | 0.98 to 1.02 | 0.96 | 0.93 to 0.98** | 0.97 | 0.94 to 0.99* | 1.21 | 1.18 to 1.25** | 1.22 | 1.19 to 1.26** | 1.16 | 1.13 to 1.20** | 1.18 | 1.15 to 1.22** |
| **Mother's nationality** | | | | | | | | | | | | | | | | |
| Locals | 1 | | 1 | | 1 | | 1 | | 1 | | 1 | | 1 | | 1 | |
| Foreigners | 0.68 | 0.66 to 0.71** | 0.59 | 0.57 to 0.62** | 0.73 | 0.70 to 0.76** | 0.65 | 0.62 to 0.68** | 0.75 | 0.72 to 0.79** | 0.65 | 0.62 to 0.69** | 0.82 | 0.78 to 0.86** | 0.72 | 0.68 to 0.76** |
| **Mother's socioeconomic status** | | | | | | | | | | | | | | | | |
| Disadvantaged population | 1.56 | 1.48 to 1.63** | 1.60 | 1.52 to 1.68** | | | 1.42 | 1.36 to 1.50** | 1.63 | 1.54 to 1.72** | 1.67 | 1.58 to 1.77** | | | 1.48 | 1.40 to 1.56** |
| Annual income <18.000 | 1 | | 1 | | | | 1 | | 1 | | 1 | | | | 1 | |
| Annual income 18.000–100.000 | 0.72 | 0.70 to 0.74** | 0.68 | 0.66 to 0.70** | | | 0.73 | 0.71 to 0.75** | 0.67 | 0.65 to 0.69** | 0.64 | 0.61 to 0.66** | | | 0.68 | 0.66 to 0.70** |
| Annual income >100.000 | 0.39 | 0.32 to 0.47** | 0.37 | 0.31 to 0.44** | | | 0.43 | 0.36 to 0.52** | 0.36 | 0.28 to 0.46** | 0.34 | 0.26 to 0.43** | | | 0.40 | 0.31 to 0.51** |
| **Mental disorders in the mother** | | | | | | | | | | | | | | | | |
| No | 1 | | | | 1 | | 1 | | 1 | | | | 1 | | 1 | |

(*Continued*)

**Table 3.** (*Continued*)

| | Boys (n(%)) | | | | | | | | Girls (n(%)) | | | | | | | |
|---|---|---|---|---|---|---|---|---|---|---|---|---|---|---|---|---|
| | Crude odds ratio | | Model 1 | | Model 2 | | Model 3 | | Crude odds ratio | | Model 1 | | Model 2 | | Model 3 | |
| | ORc | 95%CI | ORa | 95%CI | ORa | 95%CI | ORa | 95%CI | ORc | 95%CI | ORa | 95%CI | ORa | 95%CI | ORa | 95%CI |
| Yes | 2.30 | 2.25 to 2.36** | | | 2.28 | 2.23 to 2.34** | 2.17 | 2.12 to 2.23** | 2.40 | 2.33 to 2.47** | | | 2.37 | 2.30 to 2.44** | 2.23 | 2.17 to 2.30** |

Model 1 adjusts for age, parent's nationality and socioeconomic status. Model 2 adjusts for age, parent's nationality and mental disorders in the parent. Model 3 adjusts for all of the variables.

ORc: Crude Odds Ratio; ORa: Adjusted Odds Ratio; CI: Confidence Interval.

** p-value < 0.001

* p-value < 0.05.

Disadvantaged population: People who received an economical subsidy and those exempt from pharmaceutical co-payment.

2.30, respectively) compared with having a father with MD (OR 1.66, 95%CI 1.61–1.70 and OR 1.72, 95%CI 1.66–1.77, respectively).

A gradient of risk by SES was observed. In the most disadvantaged population, there was an elevated risk of suffering MD when compared with the subsequent more privileged SES. This higher risk was more accentuated when the low SES was the mother's (OR 1.42, 95%CI 1.36–1.50 for boys; OR 1.48, 95%CI 1.40–1.56 for girls), than when it was the father's (OR 1.21, 95%CI 1.16–1.27 for boys; OR 1.17, 95%CI 1.11–1.24 for girls).

## Discussion

The results of this study support the hypothesis that the exposure of children to adverse life conditions related with having a parent with MD increases the odds of presenting MD themselves. This is consistent with the results found in previous research, as one of the most consistent and potent non-specific risk factor for the development of MD in children is a parental history of MD [5, 45]. The underlying mechanism that could explain how parental MD can increase the risk of the offspring's MD is likely to be complex and multifactorial, including both genetic and environmental factors. Existing literature has evidenced the genetic heritability of some MD [63] but also the importance of gene-environment interactions in the intergenerational transmission of MD [15, 18]. The adverse environment for the children derived from the presence of MD in the parent can come from the intercorrelation between parental MD and its effect through parenting abilities, cognitions, behaviours, affect and the capacity for quality paternal interactions [64].

Furthermore, this study found that the intergenerational transmission of MD from mothers to offspring had a higher association than the one from fathers to offspring, as previous research suggested [50–53]. This fact could be related with both, social and biological factors. Parental education is determinant for the healthy development of children [6]. As socially constructed gender roles place the woman as the principal caregiver, they may be the main familiar educational influence of the children. Healthy mother-infant interactions are critical to the development of healthy children and the quality of maternal caregiving behaviour during infancy has been linked to risk for psychopathology [65]. Heritable genetic factors of some MD [66] and intrauterine exposures to MD [65] could be also playing a role in explaining some of these differences.

Although a greater role of the mental health of the mother has been supported by the existing literature, it is also true that typically, studies have focused on the mother's mental health,

neglecting the importance of the father's mental health. However, research has shown that paternal MD are also associated with MD in their children [67]. Moreover, research has found evidence that a good paternal mental health may buffer the influence of a mother's poorer mental health on a child's behavioural and emotional problems [68].

When adjusting for all variables, the association between the different SES and the presence of MD is also consistent with previous research. Those lower on the socioeconomical hierarchy are more likely to have MD, maybe due to their greater exposure to stressful events throughout life: less favorable economic, social, and environmental conditions and less access to buffers and supports [69]. The mother's SES having a more pronounced effect could be explained also by the fundamental importance of the primary caregiver in the early years. The mother's educational level, highly related with the level of income [70], has been found greatly associated to the health of their children [71].

As for the other adjusting variables, older ages (11 years old and more) resulted a protective factor in boys but a risk factor for girls. This could be related with the different age of onset of the disorders they tend to suffer, as girls have greater rates of mood and anxiety disorders, and boys have greater rates of behaviour disorders [5], which have earlier onsets than mood disorders [72]. Previous studies have found that MD that have a marked male preponderance such as autism, developmental language disorders, attention deficit disorder with hyperactivity, and dyslexia, have an earlier onset than emotional disorders, such as depressive conditions and eating disorders, which show a marked female preponderance and an adolescent-onset [73]. Also, boys and girls may differentially experience interactions of environmental influences [74] such as the effects of the exposure to a parent with a MD on their predisposition to suffer MD. Their relationship with the parents may also vary across both genders and children's age ranges, modulating the potential effect of parental psychopathology on children's mental health [75].

While previous studies found the migration process to be a risk factor [76], in this study there were less odds of presenting MD when the parent was a foreigner. This could be related with the way MD are approached and the fact that foreigners show a lower use of health services than locals [77]. The "healthy migrant effect" could be also playing a role [78].

## Limitations and strengths

This study has some limitations. First, having anonymized data, it cannot be known exactly if the HCH is a parent or not, although it is known from the operation of the registry that this will be the predominant case. However, even if they are not the direct parents of the child, the minors will always be in charge of the insured person, and thus, exercising the role of legal tutor. Hence, whereas the genetic factors will not be playing a role, the environmental factors hypothesized as contributors to the associations will also be applicable.

The presence of MD is approached by the use of the health care services, but it is not possible to determine the degree of need behind this use. Therefore, it can only approximate the real state of MD of the target population. However, it has been found that need is the most important predictor of the probability of using any of the health care services [79].

Also important to note, this study only considered the presence of MD in the parents during the same target year, not assessing the possible existence of MD of the parents during the whole life of the children.

The variable of SES is based on the pharmacy copay that each individual has according to his/her income tax declaration. Therefore, it measures individual income. However, in those cases where couples are taxed together, it will reflect the joint rental amount. In addition, the data in which this variable relies, does not allow a more detailed disaggregation of socioeconomic categories, which could be especially relevant in the 18.000–100.000 category.

The study relies on an exclusive analysis of data of public health services and there may be an influence of double coverage in the SES gradients. Thus, a loss of information due to the use of private services cannot be discarded. This potential bias will occur mainly in the group with the highest SES and could occur, to a lesser extent, in the second group with the highest SES.

In acknowledging these limitations, this study has also several strengths. First, most of the previous research that has analyzed inequalities in health uses aggregate data or data from population samples, which have limitations in terms of being truly population representative. The potentiality of this study is the analysis of individual data of the whole population of Catalonia of the targeted age and year. To our knowledge, this is the first study about this subject conducted with a sample of these characteristics that has linked children with their parents.

Also remarkable, the data in which this study is based comes from administrative records of the public health system and thus, implies no extra economic cost for the collection of data, as data from medical records are repurposed. Linked administrative databases offer a powerful resource for public health research [80, 81].

While one large limitation of administrative data is the frequent lack of individual-level SES information [80], this is not the case of this study: the pharmaceutical co-payment provides an individual measure, entailing a great strength.

Another strength is that parents and children can be linked through administrative data. In addition, the possibility of analyzing separately the potential role of the father or mother's MD and SES, is a meaningful strength. Moreover, most of the previous research determining that worse mental health of the child is associated with low SES or paternal MD, has treated these two factors separately. This study has assessed both together, also including gender approach.

## Conclusions

This study points out the existence of an association between the presence of MD in parents and the presence of these problematics in their children. It also highlights a greater role of the mental health of the mother than the one from the father. An association between SES of the parent and MD in children has also been found, placing those children in a vulnerable socio-economic position at greater risk of having MD. The influence of the SES is higher when it is the status of the mother than of the father.

The findings of this study offer a preliminary indication that policies aiming to reduce the intergenerational transmission of MD should target both socioeconomic inequalities and the causes behind parental MD, which have been seen to include a high environmental component. The findings also point to the importance of addressing gender differences in these associations. Gender roles intersect with critical structural determinants of health and so, incorporating a gender approach is essential when assessing MD and their intergenerational transmission.

## Supporting information

**S1 Dataset.**
(XLSX)

## Acknowledgments

No financial support was received for this work. The authors thank all the helpful comments of the following: Garazi Carrillo Aguirre, Cristina Colls Guerra, Raúl González Rodríguez and Neus Carrilero Carrió. The authors have declared that they have no competing or potential conflicts of interest.

## Author Contributions

**Conceptualization:** Alba Oliver-Parra, Albert Dalmau-Bueno, Dolores Ruiz-Muñoz, Anna García-Altés.

**Data curation:** Alba Oliver-Parra, Albert Dalmau-Bueno.

**Formal analysis:** Alba Oliver-Parra, Albert Dalmau-Bueno.

**Investigation:** Alba Oliver-Parra.

**Methodology:** Alba Oliver-Parra, Albert Dalmau-Bueno, Dolores Ruiz-Muñoz.

**Supervision:** Albert Dalmau-Bueno, Dolores Ruiz-Muñoz, Anna García-Altés.

**Validation:** Albert Dalmau-Bueno, Dolores Ruiz-Muñoz, Anna García-Altés.

**Visualization:** Alba Oliver-Parra.

**Writing – original draft:** Alba Oliver-Parra.

**Writing – review & editing:** Albert Dalmau-Bueno, Dolores Ruiz-Muñoz, Anna García-Altés.

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
