## [Decision Letter · Decision Letter 0]

11 Jun 2020

PONE-D-20-07592

A nationwide, register-based study: To what extent are parents’ mental disorders and socioeconomic status associated with children’s mental disorders?

PLOS ONE

Dear Author,

Thank you for submitting your manuscript to PLOS ONE. After careful consideration, we feel that it has merit but does not fully meet PLOS ONE’s publication criteria as it currently stands. Therefore, we invite you to submit a revised version of the manuscript that addresses the points raised during the review process.

We look forward to receiving your revised manuscript.

Kind regards,

Luca Cerniglia, PhD

Academic Editor

PLOS ONE

Journal Requirements:

Reviewers' comments:

Reviewer's Responses to Questions

**Comments to the Author**

1. Is the manuscript technically sound, and do the data support the conclusions?

Reviewer #1: No

Reviewer #2: Partly

2. Has the statistical analysis been performed appropriately and rigorously? 

Reviewer #1: Yes

Reviewer #2: Yes

3. Have the authors made all data underlying the findings in their manuscript fully available?

Reviewer #1: Yes

Reviewer #2: Yes

4. Is the manuscript presented in an intelligible fashion and written in standard English?

Reviewer #1: Yes

Reviewer #2: Yes

5. Review Comments to the Author

Reviewer #1: Dear editor,

Thank you very much for the invite to review the manuscript entitled “A nationwide, register-based study: To what extent are parents’ mental disorders and socioeconomic status associated with children’s mental disorders?”.

I read with very much interest the paper, that is focused on a topic of growing importance in the clinic and research. Actually, in fact, mental disorders represent a very relevant issue in the international scientific literature. In particular, many researchers have concentrated their interest on the link between parental psychopathologies and offspring’s maladaptive outcome.

My overall impression on this manuscript is positive.

The authors, in fact, in their work discuss thoroughly the theme of the relationship between parental social and psychological features and offspring’s well-being. In particular, the authors emphasize in the paper the importance of the role of maternal and/or paternal mental disorder, together with low socioeconomic levels, on children’s psychological outcome.

The writing is overall understandable and the study appears to be sound (form and contents are quite clear). The introduction section, the general aim and results are clearly recognized. Moreover, the use of written English is quite good and clear.

These elements as a whole represents a manuscript’s strengths.

However, I would like to comment on some points in the study, so that the authors can improve the final version of their work.

Title

The title in the full version is perhaps too long and lengthy. I suggest to indicate the core theme of the study (relationship between parental characteristic and offspring’s psychological outcome), without specifying the extension of the possible association between the variables.

I also suggest the authors to indicate the method of study used, if possible, in a simpler way (replacing the expression “A nationwide, register-based study”).

In the short title (“Parent’s characteristics impact on child’s psychopathology”), and in other part of the paper, the word impact refers to the evaluation of the possible influence of parental conditions on children’s mental health that cannot be properly detected through a retrospective study.

Introduction

In the introduction (line 41-56) the authors should add epidemiological data on gender differences in mental health, also in relation to the age, which represent an element of attention in data collection of this paper. Moreover, the authors talk about subjects belonging to different age groups and gender in little detail (woman, girls, children, adolescent).

The authors should try to make to readers more understandable the part in which the research is explained and in which the gender approach is presented between the methodological choices.

This is because, more in particular, the scientific literature highlighted the relevance of internalizing and externalizing symptomatology in children and adolescents belonging to different genders and age groups.

Likewise, it may be helpful to better clarify (line 57-64) the aspect of family environment and offspring’s psychopathological outcome with more reference to empirical study on children and adolescents separately.

For these purposes I indicate some recent international studies to be consulted.

-Cerniglia L, Cimino S, Erriu M, Jezek S, AlmenaraCA, Tambelli R (2018) Trajectories of aggressive and depressive symptoms in male and female overweight children: Do they share a common path or do they follow different routes? PLoS ONE 13(1):e0190731.https://doi.org/ 10.1371/journal.pone.0190731

-Davis, S., Votruba-Drzal, E. & Silk, J. S. Trajectories of Internalizing Symptoms From Early Childhood to Adolescence: Associations With Temperament and Parenting. Soc Dev. 24, 501–520 (2015).

-Cimino, S., Cerniglia, L., & Paciello, M. (2015). Mothers with depression, anxiety or eating disorders: Outcomes on their children and the role of paternal psychological profiles. Child Psychiatry & Human Development, 46, 228–236. doi:10.1007/s10578-014-0462-6

-Boursiquot, P. E. et al. Emotional dysregulation in children: the impact of prenatal stress and maternal sensitivity. Psychiatry 59, 9, 497–508 (2014).

-Noh, J. W., Kim, Y. E., Park, J., Oh, I. H. & Kwon, Y. D. Impact of parental socioeconomic status on childhood and adolescent overweight and underweight in Korea. J Epidemiol. 24, 3, 221–229 (2014).

In the proposal of the general purpose of the research, the justification for the aim could be made clearer (line 74-78): the collect of whole population data to analyzing the variables of interest (mental disorder and social status) is very useful in comparison with previous studies on representative samples, but since the investigation is conducted as retrospective study , it would be appropriate to reformulate the whole expressions (the use of predictor term for example).

Method

The section on methodology could be improved in the choice of titles to be given to subsections.

A possible articulation could be the following:

Research Methods

-Subjects and procedure

-Measures

-Statistical analysis

The authors should not list the variables but include them within specific objectives and hypotheses of the study to be articulated with respect to the general purpose (about gender, age, genetic aspects, social and economical elements).

I didn’t find explicit indication about the authorization by the ethic and scientific committees in charge of the study.

The vertical format of table 1 results too long. If is possible it would be useful to separate the data about Father as health card holder and Mother as health card holder, also with a short title.

Discussion and conclusion

In the section Discussion (of the results), authors stated that “The results of this study support the hypothesis that the exposure of children to adverse life conditions derived from having a parent with MD increases the odds of presenting MD themselves”.

These results do not correspond to a starting main hypothesis formulated in a detailed and articulated way, with both general and specific objectives which it would be useful to define with more details.

As indicated above, also the use of the term is very important for the content of the paper: in this sense, the term derived suggests a causal relationship that cannot be assessed in this type of research.

The sub title Statement of principal findings could be deleted.

In the section Conclusion is highlighted “a greater role of the mental health of the mother”. I suggest to insert some references to studies on the role of the father figure in the well-being of children and adolescents, as empirical international literature supports (possible role of mediation?).

For these purposes I indicate some recent international studies to be consulted.

-Cimino, S., Cerniglia, L., & Paciello, M. (2015). Mothers with depression, anxiety or eating disorders: Outcomes on their children and the role of paternal psychological profiles. Child Psychiatry & Human Development, 46(2), 228-236.

-Keown, L. J. (2012). Predictors of boys’ ADHD symptoms from early to middle childhood: The role of father–child and mother–child interactions. Journal of abnormal child psychology, 40(4), 569-581.

Reviewer #2: I read with interest the manuscript titled “a nationwide, register-based study: to what extent are parents’ mental disorders and socioeconomic status associated with children’s mental disorders?”. I think that the paper focuses on interesting aspects, but there were some limitations. So i think that it can be published in this journal, but with minor revision. Please find below some comments.

INTRODUCTION

The introduction is well structured and is specifically focused of the intergenerational transmission of mental disorder from parents to children, considering the possible role played by socioeconomic status and parental gender.

However, the theoretical model on which the authors have based the definition of “children” and “adolescents” is not explained. Childhood and adolescence represent two different development stages, with specific characteristics, risk and opportunity. Moreover, the influence of the parent's gender (mother vs. Father) has a different impact depending on the specific evolutionary phase of the child. Likewise, the gender of the child also plays a role with respect to the gender of the parent. The literature cited does not take into account this difference and the most recent scientific contributions. From the outset, theoretical framework should be clear. In fact, later in the text, there are some of them mentioned (intergenerational transmission of md; the dynamic interplay bwteen different risk factors, etc.) But the authors should, however, describe from the beginning of the theoretical framework from which they start for their own study. I suggest to see the work’s in the field of the developmental psychopathology. For example, the study by:

- Sroufe, l. A., & rutter, m. (1984). The domain of developmental psychopathology. Child development, 17-29.

- Wenar, c., & kerig, p. (2000). Developmental psychopathology: from infancy through adolescence. Mcgraw-hill.

- Cents, r. (2016). Like mother, like child?: intergenerational transmission of psychopathology; a focus on genes and parenting. John wiley & sons: new york.

The authors highlight the role played respectively by genetic and environmental influences on the development of md. However, recent evidence in the field of gene-environmet interaction, on general populations, have shown that the genetic characteristics of the child can moderate the effects of family environmental exposure (parental psychopathological risk) on children's psychopathological symptoms. Furthermore, it was highlighted that epigenetic mechanisms may be further responsible for the intergenerational transmission of psychopathological risk. I suggest citing this literature:

- Cimino, s., cerniglia, l., ballarotto, g., marzilli, e., pascale, e., d’addario, c., ... & tambelli, r. (2018). Dna methylation at the dat promoter and risk for psychopathology: intergenerational transmission between school-age youths and their parents in a community sample. Frontiers in psychiatry, 8, 303.

- Cimino, s., cerniglia, l., ballarotto, g., marzilli, e., pascale, e., d’addario, c., ... & tambelli, r. (2019). Children’s dat1 polymorphism moderates the relationship between parents’ psychological profiles, children’s dat methylation, and their emotional/behavioral functioning in a normative sample. International journal of environmental research and public health, 16(14), 2567.

- Hayden, e. P., hanna, b., sheikh, h. I., laptook, r. S., kim, j., singh, s. M., & klein, d. N. (2013). Child dopamine active transporter 1 genotype and parenting: evidence for evocative gene–environment correlations. Development and psychopathology, 25(1), 163-173.

The last part of the introduction, concerning the aims and hypotheses of the study, is very poor and should be better organized. The authors should describe the main aims of the study, reporting for each of them the main hypotheses. Based on which previous literature? I suggest citing also here the results of previous studies based on which the authors have defined their hypotheses, and on the basis of which theoretical perspective.

Finally, this is not a longitudinal study, so authors should clarify already in the introduction because, and based on which literature, it is possible to draw cause-effect conclusions in retrospective or cross-sectional studies.

METHODS

The authors have excluded children of less than 5 years of age. One of the reasons they bring back, is that there has been controversy about the validity of diagnosis of md in very young children. However, there are also many controversies with regard to the diagnosis of mental disorders in the developmental age, and different diagnostic systems for children and adolescents, as they represent evolutionary phases in which mental disorders manifest themselves with peculiar characteristics. On the basis of which diagnostic system have you understood the definition of mental disorder? These aspects should be clarified from the introduction.

The authors have divided the sample into two groups, based on the age of the children (6 to 10 and 11 to 15). I think it is the right methodology, based on what has been suggested previously on the specificities of the different evolutionary phases (childhood vs adolescence). However, the authors have not clarified the reason for this choice. As suggested above, since the introduction the authors have to clarify the definition of childhood and adolescence and how these different evolutionary phases can play a role with respect to the variables under study.

In the note of table 2 and table 3 is necessary to specify what ci means.

The table 3 is too big and confusing. I suggest we make a table with respect to the influence of the characteristics of the mother and one of the fathers.

DISCUSSION

As in the introduction, even discussions result poorly in respect to a theoretical prospective based on which the results of the study may be interpreted.

Moreover, one of the main interesting finding of the study, was that the older ages resulted a

Protective factor in boys but a risk factor for girls. The authors hypothesized that this may be due to the different age of onset of the disorders associated to boys and girls. However, there is a vast literature that has evidenced the role of the gender of the son respected to the gender of the parent, in the different phases of age (daughter and mother vs daughter and father; son and mother vs son and father).

I think the discussions should be enriched with these aspects.

6. PLOS authors have the option to publish the peer review history of their article (what does this mean?). If published, this will include your full peer review and any attached files.

Reviewer #1: No

Reviewer #2: No

---

## [Author Response · Author response to Decision Letter 0]

29 Sep 2020

Dear Dr. Cerniglia,

I would like to thank you for appreciating the merit of our work and for your attention. We also thank the reviewers for their time and their thorough evaluation of our paper. We are very pleased to be informed that the reviewers read the paper with interest and appreciated the relevant issue discussed within. We really appreciate the valuable comments, which have provided insights that improve the paper. Also, the study citations they provided us were very useful to address some of their comments. We tried to address all the suggestions you pointed out to improve our final work and changed the manuscript accordingly. Positive comments have not been answered, although highly appreciated.

Please find below a structured table answering all the comments thoroughly. 

Thank you very much, looking forward to hearing from you.

Yours sincerely,

Anna García-Altés, as author for correspondence

Academic Editor Author’s response Applied changes

Title

1) Please ensure that your manuscript meets PLOS ONE's style requirements, including those for file naming. Thank you for providing us the direct links to the style templates. We have checked the style requirements and followed the instructions received in the decision mail received regarding the file naming. We have also followed the affiliations formatting guideline. We have added some symbols in the authors part. We have introduced some format changes such as the indent of the text, the level 2 Headings and the citation of the reference number in square brackets. The rebuttal letter labelled “Response to Reviewers”, the marked-up copy labelled “Revised Manuscript with Track Changes”, and the unmarked version labelled “Manuscript”.

2) In your Data Availability statement, you have not specified where the minimal data set underlying the results described in your manuscript can be found. […]

Upon re-submitting your revised manuscript, please upload your study’s minimal underlying data set as either Supporting Information files or to a stable, public repository and include the relevant URLs, DOIs, or accession numbers within your revised cover letter. […] Any potentially identifying patient information must be fully anonymized.

Important: If there are ethical or legal restrictions to sharing your data publicly, please explain these restrictions in detail. […] We have introduced a “Footnotes” section including a sub-section about “Data Availability” to address this issue. Page 25, lines 450-460: “No additional data available. No additional data available. Data cannot be shared publicly because of confidentiality. Data are available from the Government of Catalonia Institutional Data Access (PADRIS Program) for researchers who meet the criteria for access to confidential data. We had access to the data as research staff of the Agència de Qualitat i Avaluació Sanitàries (AQUAS). As established by the PADRIS Program, the anonymized and unidentified data will be accessible to the research staff of accredited research centers under specific circumstances. Plos One can access aggregated data under request. You can check up-to-date regulations at http://aquas.gencat.cat/ca/ambits/analitica-dades/padris/ and address any inquiries to AQuAS director, Dr. Cesar Velasco (cesarvelasco@gencat.cat).”

3) PLOS requires an ORCID iD for the corresponding author in Editorial Manager on papers submitted after December 6th, 2016. Please ensure that you have an ORCID iD and that it is validated in Editorial Manager. To do this, go to ‘Update my Information’ (in the upper left-hand corner of the main menu), and click on the Fetch/Validate link next to the ORCID field. This will take you to the ORCID site and allow you to create a new iD or authenticate a pre-existing iD in Editorial Manager. Please see the following video for instructions on linking an ORCID iD to your Editorial Manager account: https://www.youtube.com/watch?v=_xcclfuvtxQ

The corresponding author of this manuscript has already an ORCID iD. Not applicable. 

4) Please include a separate caption for each figure in your manuscript. We only have one figure in our manuscript, which was uploaded separately and in the format required (TIFF). We have reuploaded the figure to make sure it was in the correct format. 

 

Reviewer #1 Author’s response Applied changes (new text in green)

Title

1) The title in the full version is perhaps too long and lengthy. I suggest to indicate the core theme of the study (relationship between parental characteristic and offspring’s psychological outcome), without specifying the extension of the possible association between the variables.

I also suggest the authors to indicate the method of study used, if possible, in a simpler way (replacing the expression “A nationwide, register-based study”). We agree with erasing the possible magnitude of the association from the title. We have also introduced the method of study used in the title and tried to make it simpler. Page 1, lines 1-3: replaced title. “A nationwide, register-based study: To what extent are parents’ mental disorders and socioeconomic status associated with children’s mental disorders?” changes to “Relationship between parents’ mental disorders and socioeconomic status and offspring’s psychopathology: a cross-sectional study”

2) In the short title (“Parent’s characteristics impact on child’s psychopathology”), and in other part of the paper, the word impact refers to the evaluation of the possible influence of parental conditions on children’s mental health that cannot be properly detected through a retrospective study. Thank you for pointing this out. We have changed the word “impact” for the word “role” in the short title to avoid the misleading conclusions. Page 1, line 4: “Parent’s characteristics and child’s psychopathology”

Introduction

3) In the introduction (line 41-56) the authors should add epidemiological data on gender differences in mental health, also in relation to the age, which represent an element of attention in data collection of this paper. Moreover, the authors talk about subjects belonging to different age groups and gender in little detail (woman, girls, children, adolescent). First of all, thank you for providing us suggested literature to address your comments regarding our introduction. It was very useful.

The reason behind deciding not to delve into in epidemiological data regarding MD was that we studied MD as a whole, not separating it by diagnostics. However, seeing your comment, we have provided further information regarding gender differences in MD overall. Moreover, in relation to age, a new paragraph has also been introduced describing differences in the age of onset. 

In relation with the age groups and the gender, more detailed information has also been added. Page 4, lines 57-64: “Developmental psychopathology includes consideration of sex differences and, especially, differences in the socialization of males and females. Both symptomatology and behaviour patterns predictive of later MD differ for males and females [9], placing gender as a critical determinant of mental health. Although there do not appear to be gender differences in the overall prevalence of MD [10], significant differences in the patterns and symptoms of the MD have been reported: women and girls show higher rates and lifetime risk of mood or internalizing disorders, while men and boys have a higher prevalence of behavioural or externalizing disorders [5,11–13].”

Pages 4-5, lines 65-90: “This general pattern can be seen throughout all life stages, but some differences across age groups have also been reported. Studies find that in childhood, there is a higher prevalence of conduct disorders among boys than among girls. During adolescence, girls have a much higher prevalence of depression and eating disorders than boys, while boys present more anger problems and high-risk behaviours. The median age of onset has been found much earlier for anxiety disorders and impulse-control disorders than for substance use disorders and mood disorders [7,14]. There have also been found gender differences in the age of onset of severe MD such as schizophrenia and bipolar depression, with men typically having an earlier onset of schizophrenia, while women being more likely to exhibit serious forms of bipolar depression [10].”

Pages 5-6, lines 101-112: “During childhood and until the conclusion of adolescence, important biological, psychological and emotional changes occur in human beings. During this stage, the key dimensions of health are developed: the physical, cognitive and psychosocial dimensions [26]. While there is no uniformity in the terminology used to designate the stages of childhood nor the age ranges [27–29], it is well established that development throughout the different periods of childhood influence mental health outcomes across the lifespan: the prenatal period and early childhood -approximate age range: until 5 years of age- [30], middle childhood -approximate age range: 5 to 12 or so years of age- [31,32] and also adolescence - approximate age range: 12 or so to 20 years of age, divided in early (11-14), middle (14-17) and late adolescence (17 and up)- [33–36]. During these life stages there is a greater vulnerability to the characteristics of the environment [30,37–39] due to the high malleability of biological systems [40].”

4) The authors should try to make to readers more understandable the part in which the research is explained and in which the gender approach is presented between the methodological choices.

This is because, more in particular, the scientific literature highlighted the relevance of internalizing and externalizing symptomatology in children and adolescents belonging to different genders and age groups. It has been considered a necessary practice to disaggregate all epidemiological data by sex allowing gender analysis of data. Furthermore, we have already touched the subject regarding the internalizing and externalizing disorders, but as we are considering MD as a whole.

However, understanding the relevance of the subject we have introduced some changes and extra information trying to address this petition, as can be seen across the introduction, signalling the importance of gender and also age (responding to suggestion number 3). Page 4, line 57-64: “Developmental psychopathology includes consideration of sex differences and, especially, differences in the socialization of males and females. Both symptomatology and behaviour patterns predictive of later MD differ for males and females [9], placing gender as a critical determinant of mental health. Although there do not appear to be gender differences in the overall prevalence of MD [10], significant differences in the patterns and symptoms of the MD have been reported: women and girls show higher rates and lifetime risk of mood or internalizing disorders, while men and boys have a higher prevalence of behavioural or externalizing disorders [5,11–13].”

Page 6, line 122-130: “There has been consistent evidence to demonstrate that parental MD affects almost every aspect of child development [44] and negatively affects the mental and emotional health of children [6]. Children of parents with MD present a greater risk of psychological problems [45–48]. Previous research indicates that for mothers, the association between parental and child psychopathology is specific, whereas for fathers it is non-specific: mothers’ internalizing problems have been associated to child internalizing problems and the same applied for externalizing problems [49]. Furthermore, existing research, albeit limited, suggests that maternal intergenerational transmission of MD is particularly strong [49–54].”

5) Likewise, it may be helpful to better clarify (line 57-64) the aspect of family environment and offspring’s psychopathological outcome with more reference to empirical study on children and adolescents separately. Thank you for pointing out this need for further clarification. Some references regarding each stage of life had been added, aiming to deepen into this issue. Also, a reference to the framework of developmental psychology has been added to reinforce the aspect of family environment and offspring’s psychopathological outcome. Page 6, line 113-121: “Children are strongly influenced by their environments, especially by their family unit. At this stage in life, the family is the main influence on the child's development, and the most influential characteristics of the family environment are its economic and social resources, including family members’ health, especially early in this period [30] but also in late childhood and adolescence [41,42]. Different theories in developmental psychology conceive the interaction between the caregiver and the child as crucial to psychological outcomes. Following such theories, the risk of MD in children can be seen as a result of complex interactions across multiple levels of analysis from a child’s genetic predispositions to environmental exposures such as their family conditions [41,43]. 

There has been consistent evidence to demonstrate that parental MD affects almost every aspect of child development [44] and negatively affects the mental and emotional health of children [6].”

6) In the proposal of the general purpose of the research, the justification for the aim could be made clearer (line 74-78): the collect of whole population data to analyzing the variables of interest (mental disorder and social status) is very useful in comparison with previous studies on representative samples, but since the investigation is conducted as retrospective study , it would be appropriate to reformulate the whole expressions (the use of predictor term for example). We appreciate the comment and have made efforts to clarify it: we have erased the term “predictor” and changed the paragraph to make it clearer, as suggested. We have also erased the mention to the “extent” of the association, as was suggested for the title. Page 7, line 144-148: “Most of the previous studies that have examined the relationship between parental SES and mental disorders and child mental disorders, have used representative samples rather than the whole of population data. The aim of this study is to analyze, within a gender approach, the association between parent’s MD and their SES and MD in children using data from the whole children population of Catalonia.”

Methods

7) The section on methodology could be improved in the choice of titles to be given to subsections.

A possible articulation could be the following:

Research Methods

-Subjects and procedure

-Measures

-Statistical analysis We have improved the section on methodology by applying some of your suggestions to make it less disaggregated. We agree that some of the sub-titles could be improved. We have erased the subsection of “Data collection” and included it in the first subsection and changed the title “Variables” to “Measures” as suggested. However, we believe that it makes it simpler to maintain the separation between “Subjects” and “Procedure”, and thus, maintained this separation. Pages 8-10, lines 173, 192 and 214: the sub-sections of the methodology changed from “Study design, Data collection, Population, Variables and Statistical analysis” to “Study design and setting, Study population, Measures and Statistical analysis”. 

8) The authors should not list the variables but include them within specific objectives and hypotheses of the study to be articulated with respect to the general purpose (about gender, age, genetic aspects, social and economical elements). The objective of this section was not to list the variables per se, but to explain how they are measured and constructed. We had already defined which was the main outcome, the exposure variables and the stratification ones, and the variables treated as potential confounders, as the Strobe Checklist suggests. Also, as we have noticed revising literature, listing the variables is not an uncommon way of proceeding in the scientific papers. However, while we believe this relation you ask for is clear when reading all the paper, we have introduced some minor changes trying to respond to your request regarding some variables that could have been less explained. Page 10, line 230-234: “To address the role of gender, sex of both the children and their parents were treated as stratification variables. Age of the children was divided into to two age groups with a 5-year interval in order to adjust for the different developmental stages: 6 to 10 and 11 to 15. Nationality of the parent was dichotomized into locals or foreigners. Age and nationality were treated as potential confounders.”

9) I didn’t find explicit indication about the authorization by the ethic and scientific committees in charge of the study. We have added a “Footnotes” section including a sub-section called “Ethics approval” with some specifications regarding this subject. Page 25, line 433-449: “This study adheres to the PADRIS Program. This Program is a data analytics program for health research and innovation of the Catalan healthcare administration. It has the mission of making available to the scientific community the related health data to promote research, innovation and evaluation in health through access to the reuse and crossing of health data generated by the comprehensive National Health System of Catalonia. It ensures the use of the data goes in accordance with the legal and regulatory framework, the ethical principles and transparency towards the citizens of the program. We have had access to the data as workers within the Catalan healthcare administration under the fulfilment of the established criteria to guarantee the respect to the ethical, security and risk analysis principles of the PADRIS Program, including the anonymization of all data made available to researchers. Thus, the study did not involve any data collection, requiring neither human participants nor patient consent. For that reason, and due to the use of existing anonymised data for research, the study was exempt from institutional review committee approval. It is the standard way of proceeding in the healthcare administration to reuse the data provided by the healthcare registers for scientific purposes and to systematically check health outcomes of our population in our context.”

10) The vertical format of table 1 results too long. If is possible it would be useful to separate the data about Father as health card holder and Mother as health card holder, also with a short title. The table has been separated into two parts: children with father as health card holder and children with mother as health card holder. Table 3 has also been separated as suggested by reviewer 2 and thus, table 2 has also been modified to adhere to the formatting of the rest of the tables. Pages 11-13: Table 1 has been separated into “Table 1” and “Table 1 (continued)”.

Discussion and conclusions

11) In the section Discussion (of the results), authors stated that “The results of this study support the hypothesis that the exposure of children to adverse life conditions derived from having a parent with MD increases the odds of presenting MD themselves”.

These results do not correspond to a starting main hypothesis formulated in a detailed and articulated way, with both general and specific objectives which it would be useful to define with more details. You are absolutely right, and we have addressed this issue by detailing not only the main aim in the introduction but also the objectives and hypothesis we had. Page 7, line 146-157: “The aim of this study is to analyse, within a gender approach, the association between parent’s MD and SES and the prevalence of MD in their offspring using data from population registers of Catalonia. The specific objectives include the study of: 1) the association between parental MD and offspring’s MD; 2) the association between parental SES and offspring’s MD; and 3) the role played by the gender of both the parent and their offspring in these associations. We hypothesized that MD would be more prevalent in children with a parent with MD and with low SES. We also hypothesized that these associations would be higher in the case of maternal MD and low SES because socially constructed gender roles often place the woman as the principal caregiver. Finally, we hypothesized that there could be differences in the associations between parental characteristics and children's psychological outcome regarding the gender of the child.”

12) As indicated above, also the use of the term is very important for the content of the paper: in this sense, the term derived suggests a causal relationship that cannot be assessed in this type of research. We have replaced the term “derived” with the term “related” to eliminate the misleading suggestion of causal relationship. Page 19, line 303-305: The results of this study support the hypothesis that the exposure of children to adverse life conditions related with having a parent with MD increases the odds of presenting MD themselves.

13) The sub title Statement of principal findings could be deleted. We agree that it does not provide relevant information nor clarification. Page 19: The subtitle “Statement of principal findings” has been deleted.

14) In the section Conclusion is highlighted “a greater role of the mental health of the mother”. I suggest to insert some references to studies on the role of the father figure in the well-being of children and adolescents, as empirical international literature supports (possible role of mediation?). We find this suggestion very appropriate as we would not like the discussion to drive to the conclusion that mother’s mental health requires more attention than the father’s. We have introduced a paragraph regarding this subject and the importance of the father figure in the well-being of children. However, if you think that further explanation is needed, we would appreciate it if you could share with us some of the references for further detail. Page 20, line 328-333: “Although a greater role of the mental health of the mother has been supported by the existing literature, it is also true that typically, studies have focused on the mother’s mental health, neglecting the importance of the father’s mental health. However, research has shown that paternal MD are also associated with MD in their children [67]. Moreover, research has found evidence that a good paternal mental health may buffer the influence of a mother’s poorer mental health on a child’s behavioural and emotional problems [68].”

 

Reviewer #2 Author’s response Applied changes (new text in green)

Introduction

1) The theoretical model on which the authors have based the definition of “children” and “adolescents” is not explained. Childhood and adolescence represent two different development stages, with specific characteristics, risk and opportunity. Moreover, the influence of the parent's gender (mother vs. Father) has a different impact depending on the specific evolutionary phase of the child. Likewise, the gender of the child also plays a role with respect to the gender of the parent. The literature cited does not take into account this difference and the most recent scientific contributions. From the outset, theoretical framework should be clear. In fact, later in the text, there are some of them mentioned (intergenerational transmission of md; the dynamic interplay between different risk factors, etc.) But the authors should, however, describe from the beginning of the theoretical framework from which they start for their own study. I suggest to see the work’s in the field of the developmental psychopathology. We had a lot of debate regarding the definitions of children and adolescents as there are different biological, psychological, and cultural time scales that define the stages of the lifespan. You had a point there stating that we did not clarify this in the manuscript, and we have tried to address it. We finally decided to understand childhood comprising both young children and adolescents seeking to simplify referring to the sample as children: age span ranging from birth to adolescence. This does not mean we do not think the distinction of the different stages of childhood is crucial when analyzing factors related with mental health: we do, and so we adjusted for the variable age. We have introduced a paragraph referring to this subject and distinguishing between three periods comprised in this definition of childhood that distinguish between the different evolutionary phases. 

Regarding the theoretical framework, a paragraph has been introduced to remark the base of the study, which relies in the principles of developmental psychology. In that sense, thank you very much for providing us suggested literature in this field, it has been very useful. Also, more information regarding the implications of the gender of the parent in the transmission of psychopathology has been included.

 Pages 5-6, lines 101-112: “During childhood and until the conclusion of adolescence, important biological, psychological and emotional changes occur in human beings. During this stage, the key dimensions of health are developed: the physical, cognitive and psychosocial dimensions [26]. While there is no uniformity in the terminology used to designate the stages of childhood nor the age ranges [27–29], it is well established that development throughout the different periods of childhood influence mental health outcomes across the lifespan: the prenatal period and early childhood -approximate age range: until 5 years of age- [30], middle childhood -approximate age range: 5 to 12 or so years of age- [31,32] and also adolescence - approximate age range: 12 or so to 20 years of age, divided in early (11-14), middle (14-17) and late adolescence (17 and up)- [33–36]. During these life stages there is a greater vulnerability to the characteristics of the environment [30,37–39] due to the high malleability of biological systems [40].”

Page 6, line 113-130: “Children are strongly influenced by their environments, especially by their family unit. At this stage in life, the family is the main influence on the child's development, and the most influential characteristics of the family environment are its economic and social resources, including family members’ health, especially early in this period [30] but also in late childhood and adolescence [41,42]. Different theories in developmental psychology conceive the interaction between the caregiver and the child as crucial to psychological outcomes. Following such theories, the risk of MD in children can be seen as a result of complex interactions across multiple levels of analysis from a child’s genetic predispositions to environmental exposures such as their family conditions [41,43]. 

There has been consistent evidence to demonstrate that parental MD affects almost every aspect of child development [44] and negatively affects the mental and emotional health of children [6]. Children of parents with MD present a greater risk of psychological problems [45–48]. Previous research indicates that for mothers, the association between parental and child psychopathology is specific, whereas for fathers it is non-specific: mothers’ internalizing problems have been associated to child internalizing problems and the same applied for externalizing problems [49]. Furthermore, existing research, albeit limited, suggests that maternal intergenerational transmission of MD is particularly strong [49–54].”

2) The authors highlight the role played respectively by genetic and environmental influences on the development of md. However, recent evidence in the field of gene-environmet interaction, on general populations, have shown that the genetic characteristics of the child can moderate the effects of family environmental exposure (parental psychopathological risk) on children's psychopathological symptoms. Furthermore, it was highlighted that epigenetic mechanisms may be further responsible for the intergenerational transmission of psychopathological risk. Thank you again for providing us suggested literature in that field. It was very interesting and useful. 

We have introduced a paragraph referring to these gene-environment interactions in the section in which we highlighted the role of genetic and environmental influences. Page 5, line 93-100: “Epigenetic mechanisms have been suggested by recent research as possible pathways through which the environment interacts with genes and produces biological responses that seem to have a role in the onset and maintaining of psychopathology [18–24]. These gene-environment interactions indicate that genetic influences on the risk of children’s psychopathology are moderated by environmental factors. Although the environment affects us throughout life, its effects are especially important during the sensitive periods of biological and brain development, which begin in the prenatal period and continue through childhood and adolescence [25].”

3) The last part of the introduction, concerning the aims and hypotheses of the study, is very poor and should be better organized. The authors should describe the main aims of the study, reporting for each of them the main hypotheses. Based on which previous literature? I suggest citing also here the results of previous studies based on which the authors have defined their hypotheses, and on the basis of which theoretical perspective. You are absolutely right, and we have addressed this issue by detailing not only the main aim in the introduction but also the objectives and hypothesis we had. However, as this paragraph is part of the introduction and in the rest of this section, we have provided all the literature and citations in which our study’s hypothesis are based (also improved with all your other interesting comments regarding the gaps it had), we have not added citations in this part. Page 7, lines 147-157: “The aim of this study is to analyse, within a gender approach, the association between parent’s MD and SES and the prevalence of MD in their offspring using data from population registers of Catalonia. The specific objectives include the study of: 1) the association between parental MD and offspring’s MD; 2) the association between parental SES and offspring’s MD; and 3) the role played by the gender of both the parent and their offspring in these associations. We hypothesized that MD would be more prevalent in children with a parent with MD and with low SES. We also hypothesized that these associations would be higher in the case of maternal MD and low SES because socially constructed gender roles often place the woman as the principal caregiver. Finally, we hypothesized that there could be differences in the associations between parental characteristics and children's psychological outcome regarding the gender of the child”

4) Finally, this is not a longitudinal study, so authors should clarify already in the introduction because, and based on which literature, it is possible to draw cause-effect conclusions in retrospective or cross-sectional studies. As far as we are aware of the inherent properties our type of study, we have, at any point, meant to talk about cause-effect relations, but about association between the variables. In any case our intention was to 

We do not think that clarification about this subject needs to be addressed in the introduction, as it is an inherent property of the cross-sectional studies. However, we have introduced the word “cross-sectional” in the title to clarify already from the beginning that this is not a longitudinal study, as you mentioned. Page 1, lines 1-3: “Relationship between parents’ mental disorders and socioeconomic status and offspring’s psychopathology: a cross-sectional study”

Methods

5) The authors have excluded children of less than 5 years of age. One of the reasons they bring back, is that there has been controversy about the validity of diagnosis of md in very young children. However, there are also many controversies with regard to the diagnosis of mental disorders in the developmental age, and different diagnostic systems for children and adolescents, as they represent evolutionary phases in which mental disorders manifest themselves with peculiar characteristics. On the basis of which diagnostic system have you understood the definition of mental disorder? These aspects should be clarified from the introduction. We had a lot of discussion regarding the controversies underlying the diagnosis of mental disorders in the developmental age, and it is one of the reasons why we did not analyse the mental disorders separately, considering the different diagnosis, but as a whole. Also, because we did find literature that supported the idea that the intergenerational transmission of mental disorders could not be disorder-specific, especially in the case of the father. Moreover, as we have already mentioned in this section, there were other reasons for excluding this specific population regarding the differences in the way of providing mental health assistance to minors aged 5 and less.

On the other hand, while we had already specified in the methodology section that we were considering the group number 5 of Mental Health of the ICD-9-CM (International Classification of Diseases- Clinical Modification) Clinical Classifications Software (CCS), we have modified the sentence and enriched our explanation to clarify the definition of mental disorder. Pages 9-10, lines 215-226: “ […] all the codes from the Clinical Classifications Software (CCS) considered within the large classification group number 5 of Mental Health of the ICD-9-CM (International Classification of Diseases- Clinical Modification) were taken into account. This group includes the following subgroups, with their respective codes: Psychosis (290-299); Neurotic disorders, personality disorders and other non-psychotic mental disorders (300-316); and Intellectual disabilities (317-319).”

6) The authors have divided the sample into two groups, based on the age of the children (6 to 10 and 11 to 15). I think it is the right methodology, based on what has been suggested previously on the specificities of the different evolutionary phases (childhood vs adolescence). However, the authors have not clarified the reason for this choice. As suggested above, since the introduction the authors have to clarify the definition of childhood and adolescence and how these different evolutionary phases can play a role with respect to the variables under study. As we previously mentioned in the response to your suggestion number 1, we had a lot of discussion regarding this subject. We have found different definitions of childhood stages based on the age. Even the WHO recognizes that other agencies may use different definitions than them. Because of that we do not find appropriate to define by age the evolutionary phases. Taking this controversy into account, we divided the sample into two groups containing the same number of age categories and this way, separating the children into two different growth stages, which are aligned with the periods considered for the onset of adolescence, which vary from 8 to 15 years old. However, following your suggestion and also in relation with your first suggestion, we developed more in detail the importance of the life stages of the different evolutionary phases in the introduction. Pages 5-6, lines 101-112: “During childhood and until the conclusion of adolescence, important biological, psychological and emotional changes occur in human beings. During this stage, the key dimensions of health are developed: the physical, cognitive and psychosocial dimensions [26]. While there is no uniformity in the terminology used to designate the stages of childhood nor the age ranges [27–29], it is well established that development throughout the different periods of childhood influence mental health outcomes across the lifespan: the prenatal period and early childhood -approximate age range: until 5 years of age- [30], middle childhood -approximate age range: 5 to 12 or so years of age- [31,32] and also adolescence - approximate age range: 12 or so to 20 years of age, divided in early (11-14), middle (14-17) and late adolescence (17 and up)- [33–36]. During these life stages there is a greater vulnerability to the characteristics of the environment [30,37–39] due to the high malleability of biological systems [40].”

Page 10, lines 229-231: “Age of the children was divided into two age groups with a 5-year interval in order to adjust for the different developmental stages: 6 to 10 and 11 to 15.”

7) In the note of table 2 and table 3 is necessary to specify what ci means. A footnote has been added to clarify what CI means. We have also noticed, thanks to this suggestion, that the ORc and ORa had neither been explained and it has also been addressed. Pages 14-15, 17-18: Tables 2 and 3: “ORc: Crude Odds Ratio; ORa: Adjusted Odds Ratio; CI: Confidence Interval”

8) The table 3 is too big and confusing. I suggest we make a table with respect to the influence of the characteristics of the mother and one of the fathers. The table has been separated into two parts: children with father as health card holder and children with mother as health card holder. Table 1 has also been separated as suggested by reviewer 1 and thus, table 2 has also been modified to adhere to the formatting of the rest of the tables. Pages 17-18: Table 3 has been separated into “Table 3” and “Table 3 (continued)”. 

Discussion

9) As in the introduction, even discussions result poorly in respect to a theoretical prospective based on which the results of the study may be interpreted. We have deepened the discussions using the suggestions provided by both reviewers. We therefore want to express our gratitude in relation to the references provided. We expect our changes will meet your suggestion. Pages 19-20, lines 303-317: “The underlying mechanism that could explain how parental MD can increase the risk of the offspring’s MD is likely to be complex and multifactorial, including both genetic and environmental factors. Existing literature has evidenced the genetic heritability of some MD [63] but also the importance of gene-environment interactions in the intergenerational transmission of MD [15,18]. The adverse environment for the children derived from the presence of MD in the parent can come from the intercorrelation between parental MD and its effect through parenting abilities, cognitions, behaviours, affect and the capacity for quality paternal interactions [64].”

10) Moreover, one of the main interesting finding of the study, was that the older ages resulted a protective factor in boys but a risk factor for girls. The authors hypothesized that this may be due to the different age of onset of the disorders associated to boys and girls. However, there is a vast literature that has evidenced the role of the gender of the son respected to the gender of the parent, in the different phases of age (daughter and mother vs daughter and father; son and mother vs son and father). First of all, we have tried to deepen into the hypothesis we had included for further clarification. In addition, we have included the new hypothesis suggested. Page 21, line 343-356: “Previous studies have found that MD that have a marked male preponderance such as autism, developmental language disorders, attention deficit disorder with hyperactivity, and dyslexia, have an earlier onset than emotional disorders, such as depressive conditions and eating disorders, which show a marked female preponderance and an adolescent-onset [73]. Also, boys and girls may differentially experience interactions of environmental influences [74] such as the effects of the exposure to a parent with a MD on their predisposition to suffer MD. Their relationship with the parents may also vary across both genders and children’s age ranges, modulating the potential effect of parental psychopathology on children’s mental health [75].”

---

## [Editor Report · Decision Letter 1]

1 Oct 2020

Relationship between parents’ mental disorders and socioeconomic status and offspring’s psychopathology: a cross-sectional study

PONE-D-20-07592R1

Dear Authors,

We’re pleased to inform you that your manuscript has been judged scientifically suitable for publication and will be formally accepted for publication once it meets all outstanding technical requirements.

Kind regards,

Luca Cerniglia, PhD

Academic Editor

PLOS ONE

Additional Editor Comments (optional):

The Authors were responsive to all points and the manuscript has greatly improved. I recommend publication in the present form.
---

## [Editor Report · Acceptance letter]

8 Oct 2020

PONE-D-20-07592R1 

Relationship between parents’ mental disorders and socioeconomic status and offspring’s psychopathology: a cross-sectional study 

Dear Dr. García-Altés:

I'm pleased to inform you that your manuscript has been deemed suitable for publication in PLOS ONE. Congratulations! Your manuscript is now with our production department. 

Kind regards, 

on behalf of

Dr. Luca Cerniglia 

Academic Editor

PLOS ONE